# Mental Fatigue Effects on the Produced Perception of Effort and Its Impact on Subsequent Physical Performances

**DOI:** 10.3390/ijerph191710973

**Published:** 2022-09-02

**Authors:** Hassen Hakim, Aymen Khemiri, Oussema Gaied Chortane, Samia Boukari, Sabri Gaied Chortane, Antonino Bianco, Santo Marsigliante, Antonino Patti, Antonella Muscella

**Affiliations:** 1Biomechanics and Bioengineering Laboratory (BMBI-UMR CNRS 7338), University of Technology of Compiègne, Centre de Recherches de Royallieu Rue Personne de Roberval, 60200 Compiègne, France; hakimhassen@yahoo.fr; 2Laboratory of Exercise Physiology and Physiopathology: From Integrated to Molecular «Biology, Medicine and Health» (UR12ES06), Faculty of Medicine, University of Sousse Ibn Jazzar Sousse, Sousse 4002, Tunisia; sabrigaied1@gmail.com; 3Research Unit (UR17JS01) “Sport Performance, Health & Society”, Higher Institute of Sport and Physical Education of Ksar-Saïd, University of La Manouba, Tunis 2010, Tunisia; aymenkha3@gmail.com (A.K.); oussama.gaeid@gmail.com (O.G.C.); samiaboukari017@gmail.com (S.B.); 4Sport and Exercise Sciences Research Unit, Department of Psychology, Educational Science and Human Movement, University of Palermo, 90133 Palermo, Italy; antonino.bianco@unipa.it (A.B.); antonino.patti01@unipa.it (A.P.); 5Department of Biological and Environmental Science and Technologies (DiSTeBA), University of Salento, 73100 Lecce, Italy; santo.marsigliante@unisalento.it

**Keywords:** power training, produced perception of effort, Stroop test, cognitive task

## Abstract

The purpose of this study was to investigate the relationship between mental fatigue induced by a demanding cognitive task and impaired physical performance in endurance due to a higher perception of effort. A total of 12 healthy adults and volunteers, who had previously practiced endurance activities for 4 to 8 h per week, performed a one-hour cognitive task involving either the process of response inhibition (Stroop task) or not (visualization of a documentary as control task), then 20 min of pedaling on a cycle ergometer at a constant perception of effort while cardio-respiratory and neuromuscular functions were measured. The Stroop task induces subjective feelings of mental fatigue (vigor: 3.92 ± 2.61; subjective workload: 58.61 ± 14.57) compared to the control task (vigor: 5.67 ± 3.26; *p* = 0.04; subjective workload: 32.5 ± 10.1; *p* = 0.005). This fatigue did not act on the produced perceived effort, self-imposed, and did not affect the cardio-respiratory or neuromuscular functions during the subsequent physical task whose type was medium-term endurance. Regardless of the mental condition, the intensity of physical effort is better controlled when the participants in physical activity control their perception of effort. Mental fatigue does not affect subsequent physical performance but estimated perceived exertion, which increases with the intensity and duration of the exercise.

## 1. Introduction

To pay tribute to Angélo Mosso [1] and his works, in 1891, on the impact of mental fatigue on a subsequent physical effort—which did not receive much echo in the scientific community of the time—researchers have recently begun to investigate the effect of mental fatigue on sport performance. Generally recognized as a complex psychophysiological phenomenon, mental fatigue is a condition of fatigue caused by an increase in the demand for cognitive activities, and it is characterized by feelings of “tiredness” and “lack of energy” [2,3]. In 2009, Marcora [4] demonstrated that mental fatigue impairs the subsequent physical performance. His results were confirmed by the works of Pageaux et al. (2014) [5], MacMahon et al. (2014) [6], Smith et al. (2016) [7], and Martin et al. (2016) [8]. These studies did not show any physiological alterations in the cardiorespiratory, neuromuscular (i.e., maximum voluntary contraction), and metabolic (i.e., blood glucose) parameters due to the negative impact of mental fatigue induced by extended cognitive effort on subsequent endurance performance. These studies, however, seem to concur that the impaired performance, when preceded by a prolonged cognitive task, is linked to an increased perception of effort [4,5,6,7,8].

Significantly, it has been well investigated that athletes’ skills often decrease in the final stages of performance, with the decline being attributed to fatigue, as well as the athletes’ decision-making skills [9]. Fatigue is a complex and varied phenomenon that is referred to as a decreased ability for maximal performance and the inability to complete a task that was once achievable within a reasonable time frame [10]. Primarily, the influence of fatigue on sports performance has been attributed to a physical fatigue from both ametabolic and a neuromuscular standpoint [11]. However, numerous authors, afterwards, have shown that mental fatigue affects subsequent physical performance [12,13,14,15,16,17]. Specifically, mental fatigue impairs cognitive performance, which has been linked to executive functions, leading to reduced attention and less accurate reactions [12,13]. It must be considered that executive functions are essential for performance in sport, being strongly correlated with numerous actions, including the ability to start and stop, monitor and change behavior, and plan for next steps [18].

Importantly, since mental fatigue affects the anterior cingulate cortex (ACC) [19], it provides a neurobiological rationale for reduced performance in a subsequent physical task. In fact, the activity of the ACC is related to the perception of effort during exercise [20]. According to the motivational intensity theory [21], these effects of mental fatigue can limit exercise tolerance regardless of any cardiorespiratory and muscle-energetic alterations.

However, fatigue-induced changes in ACC activity may also affect autonomic control [20,22] and, thus, increase cardiovascular strain as demonstrated during mental tasks [21]. Therefore, measuring motivation related to the physical tasks, perceived exertion, and the main physiological factors associated with endurance exercise performance [23,24] could be very important.

The perception of effort refers to the requirements of the task and appears as a key factor in regulating emotions and motivation [25]. It is characterized by (1) local physiological determinants (i.e., metabolic acidosis, lactatemia, availability of energy substrates, and muscle blood flow) and central ones (i.e., ventilation (VE), the volume of oxygen consumed (VO2), heart rate (HR), etc.) [16], which vary by acting on local and central sensory signals [26], and, also, (2) psychological factors [4,27]. The perception of effort is a multidimensional variable that reflects the difficulty of a task at a given moment of effort. The Borg scale is the most used to measure the rate of perceived exertion (RPE) [28]. 

In the literature, two models of perceived exertion are defined: estimation and production models [29]. The estimation model, the most used, follows an arbitrary rating methodology assigned to the RPE during the evolution of the exercise [29]. The production model is an a priori estimate of the level of requirements for the task at hand or the likelihood of its success [18]. The various previous studies that investigated the impact of mental fatigue on physical performance [4,5,6,7,8] used only maximum effort and very short duration, where the perceived exertion was used as the estimation model to explain the decline in performance and not the monitored factor being produced.

Perceived exertion seems to be a determining factor in endurance performance as well as mental fatigue, which is detrimental to information processing involved in this perception. This interaction was explained by the fact that some cognitive tasks, such as the Stroop test or AX-Continuous Performance Test (AX-CPT), induce adenosine accumulation in the anterior cingulate cortex, the area related to the perceived exertion [29,30]. These cognitive tasks create a mental fatigue that results in errors in judgment, perception, memory, and slowed reactions [30,31].

The Stroop test is based on the phenomena of associations and inhibitions (or interference). The so-called “Stroop effect” is the interference produced by irrelevant information during the execution of a cognitive task. Several theories are used to explain it: the processing speed theory (the brain reading words faster than recognizing colors), the theory of selective attention, and automaticity (color recognition not being an automatic process) [32]. Fairclough and Houston (2004) [31] demonstrated that mental effort due to response inhibition, during a prolonged duration of Stroop test, causes overconsumption of blood glucose.

In this study, the goal is to analyze the impact of a demanding cognitive task likely to cause mental fatigue on a sub-maximal exercise performance achieved at a constant perception of exertion based on the production model. Assuming that the perception of effort is the factor explaining the alteration of physical performance [4,5], we can infer that a constant generated stress perception of the performance will be impaired in the situation preceded by a prolonged cognitive task compared to a control position, and this on a sub-maximal exercise and over a relatively prolonged period. Thus, we hypothesize that the performance (i.e., power output) on sub-maximal cycling exercise with a duration of 20 min will be inferior when preceded by a prolonged Stroop task (i.e., one hour), compared with the same pedaling exercise not preceded by a cognitive bias.

## 2. Materials and Methods

### 2.1. Subjects

Twelve healthy adults (5 women and 7 men, aged between 25 and 30 years and mean = 28 ± 7.5 years, height = 175.9 ± 5.9 cm, body mass = 68.7 ± 9.1 kg), volunteer students and teachers from the UFR science and techniques of physical and sports activities (STAPS) in Reims (France), were voluntarily recruited for this study. 

The main criteria for inclusion were as follows: a 4 to 8 h a week endurance activity practice and the absence of medical contraindication regarding sub-maximal exercise on a cycle ergometer. Eligibility criteria were as follows: free from any known medical diseases, injuries, color vision deficiencies, and learning disorders and free from any medication. Subjects were randomly assigned to two groups using a random number table, having a fairly homogeneous number of males and females in each group (3 women and 3 men, and 2 women and 4 men)

We performed a sample size calculation prior to recruitment by using the G*Power software (Version 3.1.9.4, University of Kiel, Kiel, Germany). The analysis revealed that 12 subjects would be sufficient to find significant differences with an 85.88% (actual power) chance of correctly rejecting the null hypothesis.

All subjects received written instructions describing the protocol and experimental procedures but were unaware of the specific objectives of the study and the underlying hypotheses. The subjects’ informed consent was obtained in conformity with the declaration of Helsinki for the experimentation on humans. The University Hospital Farhat Hached Sousse’s ethics committee approved the study, IRB 00008931.

### 2.2. Exprimental Protocol

The subjects visited the laboratory on three different occasions. During the first visit, subjects were familiarized with the experimental procedures. Each subject performed a 10 min session of the cognitive task, and we ascertained that the mistake rate would be below 20%.

Two 5-min RPE 13 blocks were performed with a 5-min recovery time between each block in order to measure the power. The coefficient of variation should not exceed 10%.

During the second and third visits, the subjects performed either a cognitive task involving the process of response inhibition (a computerized version of the Stroop test (ST) was used in this study, which lasted an hour), called Stroop session (SS), or a control task (CT), where the inhibition response process was not involved (watching an emotionally neutral documentary in the same position in front of the same screen for an hour), called video session (VS), in randomized conditions. Before and after these aforementioned tasks, blood glucose analyses were conducted together with a measure of the maximum voluntary contraction (MVC) of the quadriceps muscle performed by a dynamometer (S 3140 Load Cell-500 kg). After the cognitive task or the control task, subjects performed 20 min of pedaling with an effort perception produced and imposed by the subjects themselves with reference to the level 13 of the Borg scale positioned in front of them all through the pedaling exercise. The heart rate (HR) was recorded continuously throughout the experiment using a heart rate monitor (Polar RS 800), and gas exchange was measured with the MetaLyzer 3B while pedaling. Heart rate, cadence, and pedaling power, as well as elapsed time were not provided to participants. 

The mood was evaluated before and after the cognitive task, as well as after the endurance exercise (Brunel Mood Scale); the workload was also evaluated after the cognitive task and after the endurance exercise (NASA-TLX), and the motivation was assessed before the endurance exercise (Intrinsic Motivation Scales). A third measure of the MVC and blood glucose was taken at the end of each session.

Each participant made three visits over a 2-week period, with a minimum of 48 h of recovery time between visits. All participants were instructed to sleep for at least 7 h, to not consume any alcoholic beverage, and not to engage in vigorous physical activity the day before each visit. Participants were also advised to avoid caffeine and nicotine for at least 3 h before visiting the laboratory.

### 2.3. Obtaining Psychological and Dynamometric Parameters

#### 2.3.1. Brunel Mood Scale

The mood was evaluated before and after the cognitive task, as well as after the endurance exercise by Brunel Mood Scale (BRUMS) [33]. This questionnaire, developed by Terry et al. [34], which is based on the Profile of Mood States, contains 24 items (e.g., angry, uncertain, miserable, tired, nervous, and energetic) divided into six respective sub-scales: anger, confusion, depression, fatigue, tension, and vigor.

#### 2.3.2. NASA Task Load Index (NASA-TLX)

NASA-TLX is a widely used subjective, multidimensional assessment tool that rates perceived workload in order to assess a task, system, or team’s effectiveness or other aspects of performance [35]. Six areas were rated after being assessed on visual analogue scales (0 to 100): mental, physical, and time demand, frustration, effort, and own performance [36]. It has been comprehensively ratified, is easily used, and is comprehensively applied in various fields, such as driving, teamwork, flying, and medicine [35].

#### 2.3.3. Intrinsic Motivation Scale (IMS)

The scale assesses individual differences in intrinsic motivation in leisure behavior. The scale is composed of 24 items. Seven numeric options are available to answer the scale questions between 1 and 7 [37].

#### 2.3.4. The Mood Questionnaire (QM)

To obtain the fatigue score, we used the Mood Questionnaire (MQ), as previously reported [34]. We asked the subjects to indicate how they felt recently and, to obtain the score concerning fatigue, we summarized the answers (from 0 to 4) of the four sub-items: “exhausted”, “knackered”, “sleepy”, and “tired” of the mood questionnaire (MQ). To obtain the score regarding vigor we added the answers (from 0 to 4) of the four sub-items: “lively”, “energetic”, “dynamic”, and “awake” of the MQ. Including these items, the questionnaire consists of 20 items on a Likert-type scale (0 = not at all, 1 = a little, 2 = moderately, 3 = a lot, 4 = extreme).

To express the values of MVC in Newton (N, forces measured in Volts), we performed a calibration test using inert suspended masses. The relationship obtained is linear: 0.0042 V = 9.81 N.

### 2.4. EMG Procedure

For the vastus medial (VM) and vastus lateralis (VL) electromyography (EMG) activity, electrode locations for muscles studied were placed on the dominant leg, defined as the preferred one for kicking a soccer ball, based on the recommendations of SENIAM [38]. The reference electrode was placed on the lateral malleolus of the same leg. Prior to placing the electrodes, the surface of the skin was shaved, abraded, and cleaned with 70 % alcohol to reduce skin impedance. Maximal voluntary isometric contractions (MVIC) were performed and used to normalize VM and VL. The MVIC of each muscle were performed for three trials of five seconds with one minute of rest between repetitions [39]. Three minutes of rest were allotted between muscles being tested [40]. To obtain the MVIC values for the VM and VL, subjects were seated on a chair in 90° of hip flexion and 60° of knee flexion and resistance placed on the distal tibia, the investigator applied force to the ankle in the direction of the knee flexion. Data were recorded at 1500 Hz with Noraxon 2400 Telemyo G2 (Noraxon Corporate, Scottsdale, AZ, USA). The signal was band-pass filtered 20–500 Hz, with a 50 Hz notch filter, and the sampling frequency was 1000 Hz.

### 2.5. Statistical Analysis

Data analysis was performed independently by three researchers and merged at each step.

All data are presented as mean ± SD. The hypotheses of normality and sphericity of the data were verified by the Shapiro–Wilk test. In cases where the data did not follow a normal distribution, the nonparametric Wilcoxon matched pairs test was used.

In case the data followed a normal distribution, *t*-test was used to evaluate the effect of the condition (inhibition vs. control) on endurance performance (pedaling power and cadence), power averaged over the first two and last two minutes, electromyographic (EMG) analysis of vastus medialis and vastus lateralis muscles during the pedaling phase, HR, VE, respiratory quotient (RER), motivation scores before endurance exercise, and NASA-TLX scores after cognitive tasks and after endurance exercise.

For the VO2, data did not follow a normal distribution law; the Wilcoxon matched pairs test was used to evaluate the effect of the condition on this parameter. For the maximum voluntary contraction (MVC), vigor, and fatigue (from the mood questionnaire), as well as blood glucose, a repeated ANOVA measure was used to analyze intra- and inter temporal time interactions. In the case of significance, post hoc tests were performed using the Holm–Bonferroni method. The Statistica software (Version 8, TIBCO, Palo Alto, CA, USA) was used to perform all these tests.

## 3. Results

### 3.1. Evolution of Mood and Motivation

The mood questionnaire did not reveal any significant changes in fatigue (Table 1) during each session (ST: F (2, 22) = 1.65; *p* = 0.21; CT: F (2, 22) = 2.65; *p* = 0.09). The mood questionnaire showed a significant change regarding vigor during each session (ST: F (2, 22) = 9.38; *p* = 0.001, CT: F (2, 22) = 6.13; *p* = 0.007). Post hoc tests indicated that the vigor score significantly decreased during the cognitive task (ST: *p* = 0.001; CT: *p* = 0.027), then significantly increased during pedaling (ST: *p* = 0.015; CT: *p* = 0.013) (Figure 1). 

The motivation questionnaire did not show a significant difference in motivation between both conditions (*p* = 0.52). A comparison according to the number of sessions in which the subject participated (Visit 2 vs. Visit 3) and independently of the cognitive task performed during these sessions also showed no significant differences (during visit 2: 24.08 ± 5.66; during visit 3: 25.16 ± 6.26; *p* = 0.35) (Figure 2).

### 3.2. Effects of the Response Inhibition on Blood Sugar and the Subjective Workload

At the end of the Stroop session, there was a significant decrease in blood glucose compared to the beginning (ST: 0.98 ± 0.17–0.85 ± 0.07; F (2, 16) = 5.79; *p* < 0.05), which was not noticed for the video session (VS: 0.91 ± 0.12–0.86 ± 0.12; F (2, 16) = 0.4; *p* > 0.05). This metabolic variable, however, did not evolve significantly following the two cognitive tasks (ST: 0.98 ± 0.17–0.93 ± 0.1; F (2, 16) = 5.79; *p* > 0.05, VS: 0.91 ± 0.12–0.88 ± 0.08; F (2, 16) = 0.4; *p* > 0.05) (Figure 3). Subjective workload data showed a significantly higher score after the Stroop task compared to the control situation (ST: 58.61 ± 14.57; CT: 32.5 ± 10.1; *p* = 0.005 by Wilcoxon); the scores after the pedaling exercise did not show any differences between both sessions (ST: 50.83 ± 15.66; CT: 48.88 ± 14.15; *p* > 0.05). Within each session, the workload evolved differently. For the Stroop session, the workload did not evolve (*p* > 0.05), while, for the control session, the workload increased (*p* = 0.012) between the questionnaire after the cognitive task (video) and the questionnaire after the pedaling exercise (Figure 4).

### 3.3. Effects of Response Inhibition on Neuromuscular Functions

For the maximum voluntary contraction (MVC) of the knee extensors, there was no evidence of any force changes within each session (ST: 451.68 ± 268.33 N–466.5 ± 215.8 N–457.09 ± 209.46 N; *p* > 0.05, VS: 473.65 ± 197.41 N–440.32 ± 225.43 N–460.67 ± 210.16 N; *p* > 0.05) (Figure 5).

The recording of the electromyography (EMG) amplitude (mean of root mean square (RMS) calculated every 2 min) of the vastus medialis and vastus lateralis during the pedaling period showed no significant differences between both conditions (vastus medial, VM: 0.147 ± 0.047–0.127 ± 0.039; *p* > 0.05, and vastus lateralis, VL: 0.105 ± 0.056–0.122 ± 0.064; *p* > 0.05) (Figure 6).

### 3.4. Effects of Response Inhibition on Endurance Exercise

The condition with the mental requirement did not reveal any significant modifications of the cadence of pedaling compared with the control situation (ST: 73.16 ± 11.47 rpm; VS: 71.75 ± 9.44 rpm; *p* > 0.05). The average pedaling power was also not impacted by the mental effort that preceded exercise and did not decline when comparing it to the condition in the absence of the cognitive requirement (ST: 143.78 ± 50.48 W; VS: 138. 91 ± 42.23 W; *p* > 0.05). When focusing on the power developed at the beginning (0–2 min) or end of the exercise (18–20 min), the statistical tests did not show any significant difference between both conditions (0–2 min: ST, 127.33 ± 50.56 W; VS 121.58 ± 40.51 W; *p* > 0.05, 18–20 min: ST: 147.87 ± 50.85 W; VS 144.93 ± 41.86 W; *p* > 0.05) (Figure 7).

### 3.5. Effects of Each Condition on Cardiorespiratory Factors during Pedaling Exercise

The heart rate did not show a significant increase during physical exertion preceded by a cognitive stress (ST: 134.16 ± 23.9 bpm–VS: 131.83 ± 20.48 bpm; *p* > 0.05). The oxygen consumption during the pedaling exercise was almost the same in both conditions (SS: 29.54 ± 10.6 mL/min/kg–VS: 29.27 ± 8.06 mL/min/kg; *p* > 0.05). Ventilation did not reveal any significant variations between the two sessions either (SS: 55.94 ± 15.27 L/min; VS: 53.47 ± 10.13 L/min; *p* > 0.05). The respiratory quotient (RER), however, was significantly greater during the pedaling exercise preceded by the response inhibition task compared to the control situation (SS: 0.92 ± 0.036; VS: 0.90 ± 0.036; *p* < 0.05) (Figure 8d).

## 4. Discussion

The aim of our study was to verify that mental fatigue increases the produced perception of the effort, which, in turn, reduces the physical performance of medium-term endurance without having any impact on the cardiorespiratory, neuromuscular, and metabolic functions. Contrary to our assumptions, the mental fatigue, which was induced by the response inhibition, did not act on the produced perception of the effort, and the subsequent physical performance was not impacted.

### 4.1. Response Inhibition and Mental Fatigue

The demanding nature of the response inhibition task was confirmed by a higher mental demand and effort evaluated by subjects in the mood and workload questionnaire, so we suggest that we managed to induce different levels of mental effort between the two conditions. Our study shows that the concentration of glucose in blood decreases regardless of the nature of the cognitive task, as Marcora et al. (2009) [4] and Pageaux et al. (2013, 2014) [5] also showed. These observations contradicted the idea of Gailliot (2008) [41], according to which blood glucose depletion is one of the underlying physiological mechanisms showing the negative effects of mental effort on subsequent physical or cognitive tasks.

### 4.2. Mental Fatigue and Neuromuscular Functions

Our experimentation showed that prolonged mental effort does not lead to a deficiency of neuromuscular function at the level of the maximum voluntary contraction of the knee extensors, which agrees with the conclusions of Pageaux et al. (2013) [5] and Rozand et al. (2014) [42]. The same applies to the activity of VM and VL muscles during the pedaling exercise, which showed no change after the demanding cognitive task. Mizuno et al. (2007) [43] and Nozaki et al. (2009) [44] suggested, however, that mental fatigue may have systemic effects, such as changes in the concentration of amino acids in the blood and could theoretically cause some peripheral fatigue. The latter was not observed in our subjects (similar increase in the demand of the motor units and the developed power).

### 4.3. Mental Effort and Subsequent Endurance Exercise

Our hypothesis was that pedaling power during sub-maximal exercise would be impaired by a demanding cognitive effort with no change in cardiorespiratory and neuromuscular function. One hour of a demanding cognitive task of response inhibition was effective to induce a state of mental fatigue, which was demonstrated by an increase in subjective feelings of fatigue and a significant reduction in cognitive performance. However, no significant effect of mental fatigue was found either on average power over the entire pedaling exercise or on average power over the first two and the last two minutes of pedaling. The hypothesis that cardiorespiratory functions would not show any significant changes has been confirmed for heart rate, oxygen consumption, and ventilation, which was not the case for the respiratory quotient, which had a significant change between the two sessions. Despite the absence of a significant difference in ventilation, the increase in respiratory quotient during the Stroop session can be explained by the decrease in the blood glucose level at the end of this session compared to the video session. The proposed exercise is not of maximum intensity, not exhaustive, and does not last long enough for the RER to rapidly tend towards 1.00. Despite this, during the Stroop session, the RER showed this trend. This is explained by the fact that the brain has an RER approaching 0.99 when it is under too much stress (indicating a predominantly carbohydrate use) and tends to increase the body’s overall RER (the total RER being the sum of the RERs of the organs and tissues requested during the exercise). 

The main conclusion of our experimental study is that mental fatigue does not have a negative effect on measured physical performance (pedaling power) nor cardiorespiratory factors (determinants of endurance performance). The motivation associated with these tasks is not affected by prior cognitive activity; therefore, mental fatigue does not alter the willingness to make efforts during the subsequent physical task, as previously demonstrated by Marcora et al. (2009) [4]. 

Concerning the exercise of pedaling and despite the difference in mental fatigue induced by the condition, the produced perception of the effort, as well as the power at the beginning of the exercise, was not affected. Then, whatever the condition, either this power was maintained or this power tended to increase (for 9 of the 12 subjects). 

An exercise requiring maximum performance and leading to exhaustion solicits the peripheral components of fatigue known to send sensory and nociceptive afferents that become accentuated with the intensity of the effort. These afferences, treated by the central nervous system previously affected by a mental fatigue, are translated into a high perception of the effort; the subject then perceives more and more the difficulty of the effort: the more the exercise persists, the more the perception of the effort increases and the performance decreases, and this is due to the cingulate cortex, which plays an important role in autonomous control during demanding cognitive and motor tests [22] and is also involved in decision making related to effort during exercise. This explains the results of previous research.

In our experiment and having proposed a medium-term endurance exercise that does not lead to exhaustion, sensory and nociceptive afferences are stable and are not treated because of peripheral dysfunction. Thus, mental fatigue did not act through the perception of the effort that was previously set (by the tests carried out during Visit 1, familiarization session) and subjects were able to manage the intensity of the self-imposed effort.

The finding is that mental fatigue, if not mediated by the perception of effort, does not directly affect subsequent physical performance and that the motor cortex can gauge the intensity of the exercise based on the sporting experience recorded there. 

Regardless of the mental conditions, the intensity of the physical effort is better controlled when the subjects control their perception of the effort.

Several limitations should be considered when interpreting the current results. First, the study sample was small due to the difficulty of recruiting large numbers of homogeneous participants. This could be an issue for the replicability of the results presented in this study and limiting the generalizability of conclusions. Therefore, it would be necessary to increase the sample size in future studies. Second, the Stroop’s task was of rather short duration compared to the duration of the cognitive tasks to which subjects are commonly exposed. Therefore, mental effort was not to be considered extreme in the present study, since higher psychobiological stress levels (i.e., mood disorders, higher than normal perception of effort during training, and reduced performance) can induce symptoms of mental fatigue even in elite athletes [45]. On the other hand, longitudinal studies on associations between mental fatigue and physical performance are scarce. It is unknown how longitudinal changes in mental fatigue are associated with changes in performance. Although, we can assume that increasing the fatigue times could induce the improvement in reaction time (for example, during the Stroop task) due to the habituation effect.

## 5. Conclusions

Our first conclusion is that mental fatigue impacts the estimated perception of the effort with the duration of the effort. On the one hand, this estimated perception appears to be the key to explaining impaired performance in situations where physical exercise is preceded by mental fatigue; on the other hand, mental fatigue does not affect the produced perception of the effort fixed beforehand to perform an endurance exercise and it does not alter the performance either.

In terms of application, it is, therefore, important to warn coaches and athletes that a mental effort involving inhibition of the response may have a detrimental effect on subsequent endurance performance, even if the subject does not feel mentally tired. Asking for the exercisers’ RPE during training may be a good indicator for the coach to monitor the intensity of the session, especially if the evaluated effort (RPE) is higher than usual for the same intensity of exercise.

## Figures and Tables

**Figure 1 ijerph-19-10973-f001:**
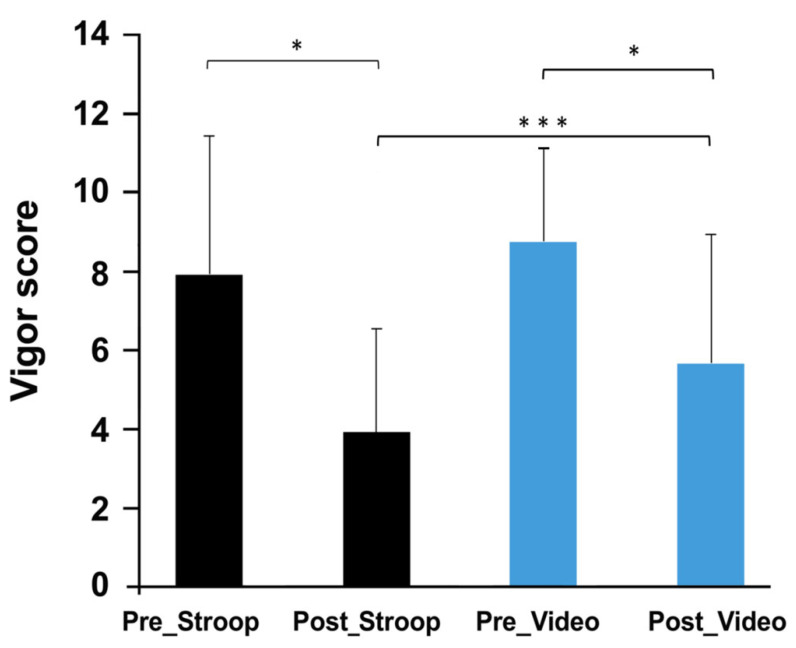
Vigor evolution evaluated during control (Pre_Stroop and Pre_Video) or during mental fatigue (Post_Stroop and Post_Video) conditions. * *p* < 0.05; *** *p* < 0.001.

**Figure 2 ijerph-19-10973-f002:**
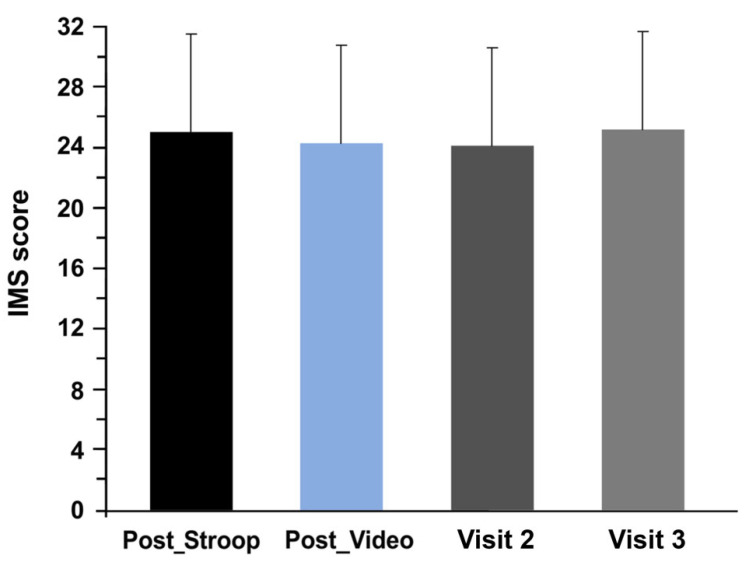
Evolution of the motivation during mental fatigue (Post_Stroop and Post_Video, in Visit 1 conditions and in the subsequent sessions in which the subject participated (Visit 2 and Visit 3).

**Figure 3 ijerph-19-10973-f003:**
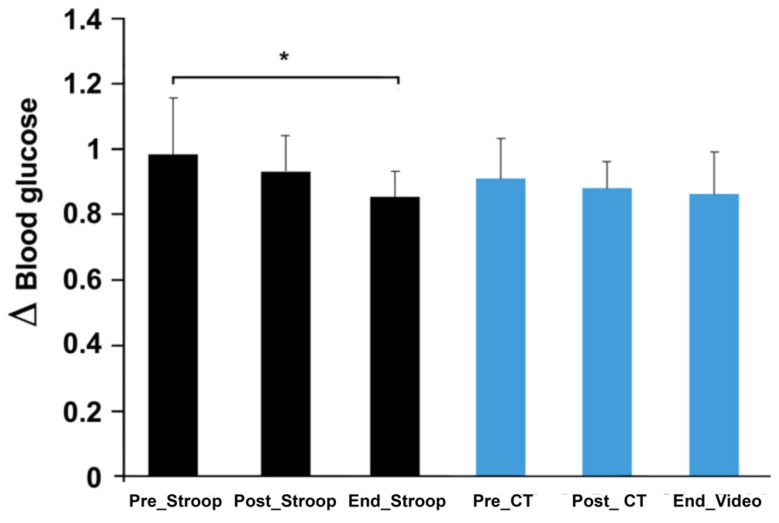
Difference in blood glucose concentration. Pre_Stroop, before Stroop task; Post-Stroop, after Stroop task; End_Stroop, at the end of Stroop session; Pre_CT, before control task; Post_CT: after control task; End_Video, at the end of video session. * *p* < 0.05.

**Figure 4 ijerph-19-10973-f004:**
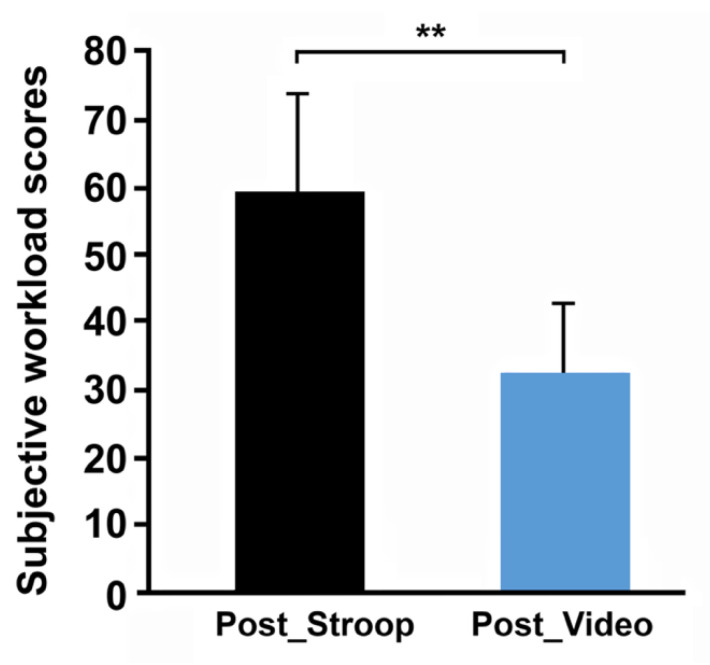
Subjective workload evaluated during mental fatigue (Post_Stroop and Post_Video) conditions. ** *p* < 0.01.

**Figure 5 ijerph-19-10973-f005:**
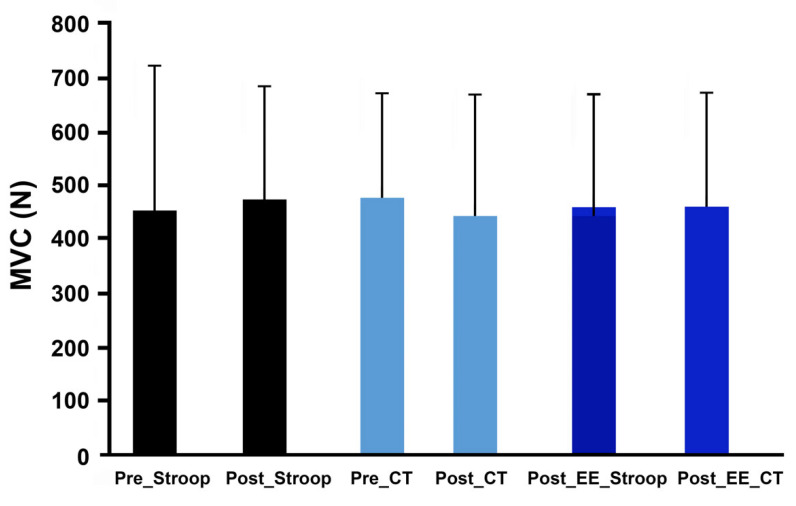
Maximal voluntary contraction (MVC) force of knee extensors measured during control (Pre_Stroop), during mental fatigue (Post_Stroop) conditions, before and after control task (Pre_CT and Post_CT, respectively), and after endurance exercise after Stoop task (Post_EE_Stroop) or after endurance exercise after control task (Post_EE_TC).

**Figure 6 ijerph-19-10973-f006:**
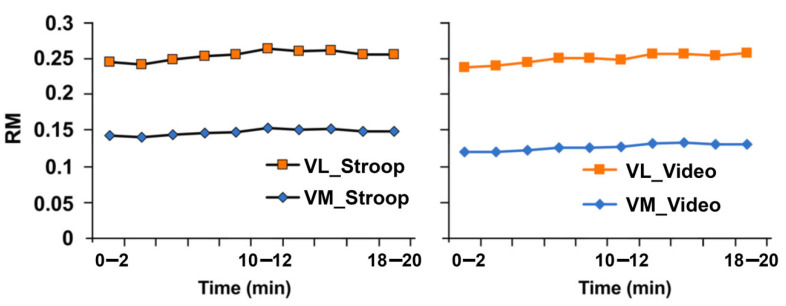
Effects of mental exertion on muscle activation, measured during endurance exercise, after Stroop or video session. RMS, root mean square; VM, vastus medialis; VL, vastus lateralis.

**Figure 7 ijerph-19-10973-f007:**
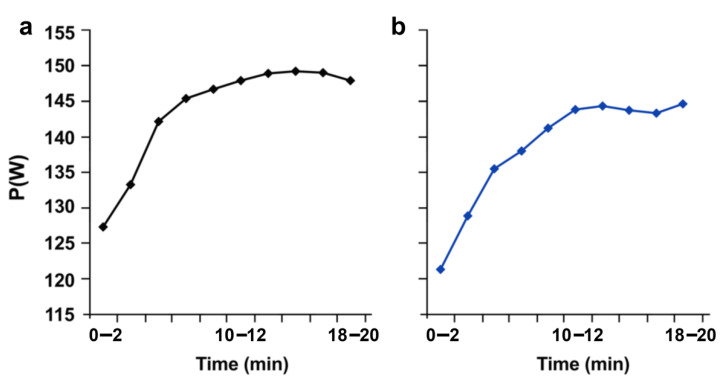
Pedaling power measured after mental fatigue. (**a**) After Stroop session; (**b**) after video session. P (W): power in watts.

**Figure 8 ijerph-19-10973-f008:**
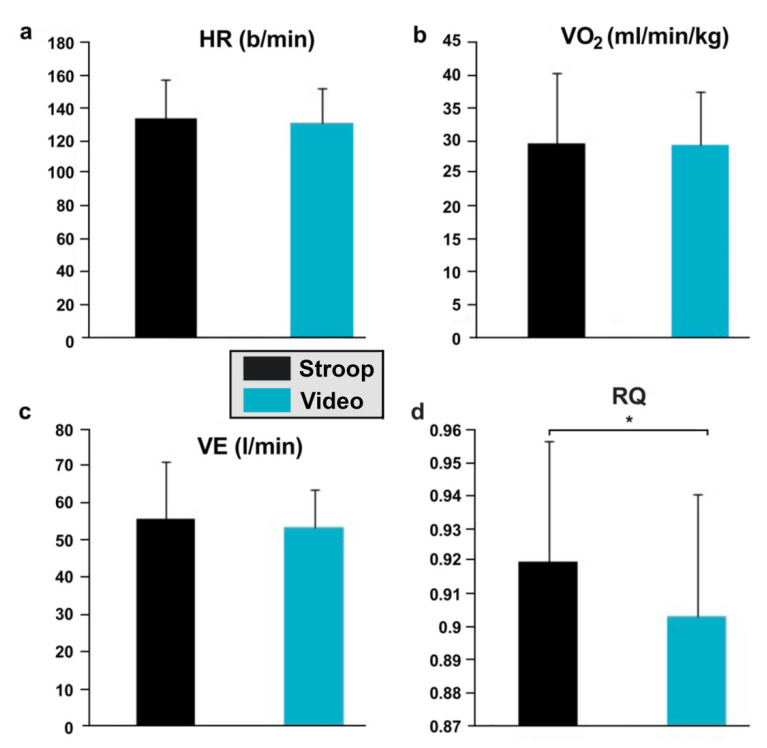
Physiological responses during endurance cycling exercise, measured after Stroop or video session. (**a**) Heart rate; (**b**) oxygen consumption; (**c**) ventilation; (**d**) respiratory exchange ratio. * *p* < 0.05.

**Table 1 ijerph-19-10973-t001:** Fatigue, vigor, and motivation scores, evaluated before (pre-cognitive) and after (post-cognitive) the cognitive task or before (pre-pedaling) and after (post-pedaling) pedaling task.

	Fatigue			Vigor			Motivation
	Pre- Cognitive	Post- Cognitive	Post- Pedaling	Pre- Cognitive	Post- Cognitive	Post- Pedaling	Pre- Pedaling
ST	2.67 ± 2.61	4.25 ± 2.86	3.42 ± 2.19	7.92 ± 3.53	3.92 ± 2.61 *	6.92 ± 4.52	25 ± 6.48
CT	3 ± 2.17	5.08 ± 2.5	3.42 ± 2.78	8.75 ± 2.38	5.67 ± 3.26	9.08 ± 3.94	24.25 ± 5.44

ST, Stroop test; CT, control task. * *p* < 0.05, significant difference between mental fatigue and control conditions. Data are presented as means ± SD.

## Data Availability

The raw data supporting the conclusions of this article will be made available by the authors without undue reservation.

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
