# Peer review of "Mental Fatigue Effects on the Produced Perception of Effort and Its Impact on Subsequent Physical Performances"

_ijerph, 2022, doi:10.3390/ijerph191710973_

Round 1
Reviewer 1 Report (New Reviewer)
First and foremost, I would like to extend my sincere congratulations to the authors for their work. The resulting paper shed light to some very interesting and enriching topics.
I consider that it can be considered a quality article but I note that the sample size may be a problem for the replicability of the results presented in this study.
On the other hand, I consider that the introductory section is an important point of an article and it should reflect well what the article is going to be, reading carefully the article, I do not consider that its writing and references are of the level required for the magazine.
Likewise, the results present the problem that since the sample is so small, these results cannot be taken into account to reproduce the article and take the results as a reference.
Finally, the bibliography should be updated with more current references on the study. There are several references below the year 2000 which I believe could be updated.
Apart from all of the above, I consider this article would be of great interest if all the points I mention were modified.
I would like to congratulate the authors for the effort made in the development of this article.
Author Response
Ref: Submission of the revised manuscript iJerph- 1838702
Mental fatigue effects on the produced perception of effort and its impact on subsequent physical performances
by Hassen Hakim, Aymen Khemiri, Oussema Gaied Chortane, Samia Boukari, Sabri Gaied Chortane, Antonino Bianco, Santo Marsigliante, Antonino Patti, and Antonella Muscella
Point-to-point reply to the Referee’s comments
Reviewer #1:
First and foremost, I would like to extend my sincere congratulations to the authors for their work. The resulting paper shed light to some very interesting and enriching topics.
I consider that it can be considered a quality article but I note that the sample size may be a problem for the replicability of the results presented in this study.
Q1.1 - On the other hand, I consider that the introductory section is an important point of an article and it should reflect well what the article is going to be, reading carefully the article, I do not consider that its writing and references are of the level required for the magazine.
R1.1 - We have refined the introduction and added other and more current bibliographic references
Q1.2 - Likewise, the results present the problem that since the sample is so small, these results cannot be taken into account to reproduce the article and take the results as a reference.
R1.2 - We are aware that our sample size may seem numerically unsatisfactory. However, we performed a sample size calculation prior recruitment by using the G*Power software (Version 3.1.9.4, University of Kiel, Kiel, Germany). The analysis revealed that 12 subjects would be sufficient to find significant differences with an 85.88% (actual power) chance of correctly rejecting the null hypothesis. This has now been specified in the Methods section.
Nevertheless, our study does have limitations related to the sample size and this was added in the conclusions.
Q1.3 - Finally, the bibliography should be updated with more current references on the study. There are several references below the year 2000 which I believe could be updated.
R1.3 - As requested, the bibliography has been comprehensively updated
Apart from all of the above, I consider this article would be of great interest if all the points I mention were modified.
I would like to congratulate the authors for the effort made in the development of this article.
Reviewer 2 Report (New Reviewer)
Thank you for inviting me to review this manuscript. It is an interesting study exploring mental fatigue on physical performance. The authors should be congratulated on conducting this experiment and using this specific research design, however to improve the manuscript I would suggest the following.
ABSTRACT
The authors here defined “physically active” the participants, I think it should be clearer that they were not only physically active but required to do endurance activities.
Line 35, authors mentioned “athlete controls his perception of effort”. Were then the participants athletes? It is not clear to which population the authors are referring and how the findings can be translated. Please make sure to be consistent with the terminology and provide a clear background on the population involved in this study.
INTRODUCTION
Authors should provide a definition of mental fatigue in the exercise and sport context, also when citing studies in lines 44-45 more background information are needed. For example, they should specify which was the physical performance and the population (adults, young adults, athletes).
Please provide more details on the link between mental fatigue and effort perception. Authors should provide clearer definition and context.
AX-CPT (AX-Continuous Performance Test), line 76. The abbreviation should be in brackets.
METHODS
Was a sample size calculation performed prior recruitment? Why 12 were included in the study? Please provide more details.
In the exclusion criteria, was age considered? Also, the authors should provide more information on how and where adults were recruited. Also, which kind of endurance activity were the participants usually doing?
Authors should provide more information on how the conditions were randomised.
Who did collect data? Was a trained researcher? Please add more information on this.
Please make sure to specify the meaning of each abbreviation. For example, line 132 EMG, electromyography is missing.
Why VM and VL were chosen to be recorded during the EMG? Authors should provide a rationale for the decision.
Please provide references for the Brunel Mood Scale, has this been validated in the population considered in the study? In general, I suggest authors to write an additional subsection to explain and provide information on the scales included in the study (Brunel, Intrinsic Motivation, workload) with supporting evidence.
It is difficult to understand section 2.3, as a lot of background information on these scales and questionnaires is missing. Please make sure to include more explanation of what is mentioned.
Was any correction of the p-value applied with the post-hoc test Holm-Bonferroni method?
Please remove direct quotes from the names of the statistical tests, these are not needed.
Please provide information on how the EMG data were filtered, analysed and sampling rate. Also, which lower limb was chosen? Dominant or non-dominant and why?
RESULTS
Table 1, please make sure to provide meaning of the abbreviations used in the table as footnotes (same for the graphs below). Please also add information on the scoring system as footnotes.
Please provide units in the graphs.
DISCUSSION
Authors should expand limitations of this study at the end of the section (i.e. longitudinal studies are needed).
Authors should include suggestions for future research.
Author Response
Ref: Submission of the revised manuscript iJerph- 1838702
Point-to-point reply to the Referee’s comments
Reviewer #2:
Thank you for inviting me to review this manuscript. It is an interesting study exploring mental fatigue on physical performance. The authors should be congratulated on conducting this experiment and using this specific research design, however to improve the manuscript I would suggest the following.
ABSTRACT
Q2.1 - The authors here defined “physically active” the participants, I think it should be clearer that they were not only physically active but required to do endurance activities.
R2.1 – The participant were healthy adults and volunteers, who had previously practiced endurance activities for 4 to 8 hours per week
Q2.2 Line 35, authors mentioned “athlete controls his perception of effort”. Were then the participants athletes? It is not clear to which population the authors are referring and how the findings can be translated. Please make sure to be consistent with the terminology and provide a clear background on the population involved in this study.
R2.2 - We checked that the terminology is consistent throughout
INTRODUCTION
Q2.3 Authors should provide a definition of mental fatigue in the exercise and sport context, also when citing studies in lines 44-45 more background information are needed. For example, they should specify which was the physical performance and the population (adults, young adults, athletes).
R2.3 – We added the following text as required by the Referee:
“Significantly, it has been well investigated that athletes’ skills often decreases in the final stages of performance, with the decline being attributed to fatigue, as well as the athletes’ decision-making skills [9]. Fatigue is a complex and varied phenomenon that is referred to as a decreased ability for maximal performance and the inability to complete a task that was once achievable within a reasonable time frame [10]. Primarily, the influence of fatigue on sports performance has been attributed to a physical fatigue both metabolic and a neuromuscular standpoint [11]. However, numerous authors, afterwards, have shown that mental fatigue affects subsequent physical performance [12-17]. Specifically, mental fatigue impairs cognitive performance, which has been linked to executive functions, leading to reduced attention and less accurate reactions [12,13]. It must be considered that executive functions are essential for performance in sport, being strongly correlated with numerous actions, including the ability to start and stop, monitor and change behavior and plan for next steps [18].”
Q2.4 Please provide more details on the link between mental fatigue and effort perception. Authors should provide clearer definition and context.
R2.4 - We added the following text as required by the Referee:
“Importantly, since mental fatigue affects the anterior cingulate cortex (ACC) (19), provides a neurobiological rationale for reduced performance in a subsequent physical task. In fact, the activity of the ACC is related to the perception of effort during exercise (20). According to the motivational intensity theory (21), these effects of mental fatigue can limit exercise tolerance regardless of any cardiorespiratory and muscle-energetic alterations.
However, fatigue-induced changes in ACC activity may also affect autonomic control (20,22) and, thus, increase cardiovascular strain as demonstrated during mental tasks (21). Therefore, measuring motivation related to the physical tasks, perceived exertion, and the main physiological factors associated with endurance exercise performance (23,24) could be very important.
R2.5 AX-CPT (AX-Continuous Performance Test), line 76. The abbreviation should be in brackets.
R2.5 Done
METHODS
Q2.5 Was a sample size calculation performed prior recruitment? Why 12 were included in the study? Please provide more details.
R2.5 - We are indeed aware that the sample size may appear small but our pilot study was also made in order to minimize the sample size of the overall study. As we stated in answer R2, a sample size calculation prior recruitment was performed by using the G*Power software revealing that 12 subjects would be sufficient to find significant differences and this has now been specified in the Methods section.
Q2.6 In the exclusion criteria, was age considered? Also, the authors should provide more information on how and where adults were recruited. Also, which kind of endurance activity were the participants usually doing?
R2.6 - Yes indeed, and the age of the participants was between 25 and 30 years. The 12 participants were volunteer students and teachers from the UFR science and techniques of physical and sports activities (STAPS) in Reims (France) and they all practiced at least 4 to 8 hours of cycling per week.
Q2.7 Authors should provide more information on how the conditions were randomised.
R2.7 Subjects were randomly assigned to two groups using a random number table, having a fairly homogeneous number of males and females in each group (3 women and 3 men and 2 women and 4 men)
Q2.8 Who did collect data? Was a trained researcher? Please add more information on this.
R2.8 Data analysis was performed independently by three researchers and merged at each step
Q2.9 Please make sure to specify the meaning of each abbreviation. For example, line 132 EMG, electromyography is missing.
R.9 - electromyography (EMG)
vastus medial, VM and vastus lateralis (VL)
Q2.10 Why VM and VL were chosen to be recorded during the EMG? Authors should provide a rationale for the decision.
R2.10 - Most researchers detect fatigue in the VL and VM muscles because it is easier to implement the electrodes as superficial muscles; they can also be controlled efficiently. The EMG can detect fatigue during dynamic and static contraction (Hayder A. Yousif et al 2019 IOP Conf. Ser.: Mater. Sci. Eng. 705 012010).
Q2.11 Please provide references for the Brunel Mood Scale, has this been validated in the population considered in the study?
R2.11 -Brandt R., Herrero D., Massetti T., Crocetta T.B., Guarnieri R., Monteiro C.B.D.M., Viana M.D.S., Bevilacqua G.G., de Abreu L.C., Andrade A. The Brunel Mood Scale rating in mental health for physically active and apparently healthy populations. Health. 2016; 8: 125-132
Q2.12 In general, I suggest authors to write an additional subsection to explain and provide information on the scales included in the study (Brunel, Intrinsic Motivation, workload) with supporting evidence.
In the text of the revised version we have added the following section:
R2.12 - Scales information
Brunel Mood Scale
The mood was evaluated before and after the cognitive task as well as after the endurance exercise by Brunel Mood Scale (BRUMS) [33]. This questionnaire, developed by Terry et al. [34], which is based on the Profile of Mood States, contains 24 items (e.g., angry, uncertain, miserable, tired, nervous, energetic) divided into six respective sub-scales: anger, confusion, depression, fatigue, tension, and vigor.
NASA Task Load Index (NASA-TLX)
NASA-TLX is a widely used subjective, multidimensional assessment tool that rates perceived workload in order to assess a task, system, or team's effectiveness or other aspects of performance [35]. Six areas were rated after being assessed on visual analogue scales (0 to 100): Mental, Physical, and Time Demand, Frustration, Effort, and Own Performance [36]. It has been comprehensively ratified, is easily used and is comprehensively applied in various fields such as driving, teamwork, flying, and medicine [35]
Intrinsic Motivation Scale (IMS)
The scale assessess individual differences in intrinsic motivation in leisure behavior. The scale is composed of 24 items. Seven numeric options are available to answer the scale questions between 1 and 7 (Duman et al., 2020).
Q2.13 It is difficult to understand section 2.3, as a lot of background information on these scales and questionnaires is missing. Please make sure to include more explanation of what is mentioned.
In the text of the revised version we have added the following section:
R2.13 - The mood questionnaire (QM)
To obtain the fatigue score, we used the Mood Questionnaire (MQ), as previously reported [27]. We asked the subjects to indicate how they felt recently and to get the score concerning fatigue, we summarized the answers (from 0 to 4) of the four sub-items: "exhausted", "knackered", "sleepy" and "tired" of the mood questionnaire (MQ). To obtain the score regarding vigor we added the answers (from 0 to 4) of the four sub-items: "lively", "energetic", "dynamic" and "awake" of the MQ. Including these items, the questionnaire consists of 20 items on a Likert-type scale (0 = not at all, 1 = a little, 2 = moderately, 3 = a lot, 4 = extreme).
Q2.14 Was any correction of the p-value applied with the post-hoc test Holm-Bonferroni method?
R2.14 – The Holm–Bonferroni method is more powerful than the classic Bonferroni correction and may be used to counteract the problem of multiple comparisons when considering several hypotheses. The correction was therefore used in order to reduce the possibility of getting a statistically significant result.
Holm, S. 1979. A simple sequential rejective multiple test procedure. Scandinavian Journal of Statistics 6:65-70
Q2.15 Please remove direct quotes from the names of the statistical tests, these are not needed.
R2.15 – Done
Q2.16 Please provide information on how the EMG data were filtered, analysed and sampling rate. Also, which lower limb was chosen? Dominant or non-dominant and why?
R2.16 – EMG procedure
For the VM and VL EMG activity, electrode locations for muscles studied were placed on the dominant leg, defined as the preferred one for kicking a soccer ball, based on the recommendations of SENIAM [38]. The reference electrode was placed on the lateral malleolus of the same leg. Prior to placing the electrodes, the surface of the skin was shaved, abraded, and cleaned with 70 % alcohol to reduce skin impedance. Maximal voluntary isometric contractions (MVIC) were performed and used to normalize VM and VL. The MVIC of each muscle were performed for three trials of five seconds with one minute of rest between repetitions [39]. Three-minutes of rest were allotted between muscles being tested [40]. To obtain the MVIC values for the VM and VL subjects were seated on a chair in 90° of hip flexion and 60° of knee flexion and resistance placed on the distal tibia, the investigator applied force to the ankle in the direction of the knee flexion. Data were recorded at 1500 Hz with Noraxon 2400 Telemyo G2 (Noraxon Corporate, Scottsdale, AZ, USA). The signal was band pass filtered 20–500 Hz, with a 50 Hz notch filter, and the sampling frequency was 1000 Hz.
RESULTS
Q2.17 Table 1, please make sure to provide meaning of the abbreviations used in the table as footnotes (same for the graphs below). Please also add information on the scoring system as footnotes.
R2.17 – Done
Q2.18 Please provide units in the graphs.
R2.18 – Done
DISCUSSION
Q2.19 Authors should expand limitations of this study at the end of the section (i.e. longitudinal studies are needed) and include suggestions for future research.
R2.19 – We added the following text as required by the Referee:
“Several limitations should be considered when interpreting the current results. First, the study sample was small due to the difficulty of recruiting large numbers of homogeneous participants. This could be an issue for the replicability of the results presented in this study and limiting the generalizability of conclusions. Therefore, it would be necessary to increase the sample size in future studies. Second, it is that Stroop's task was of rather short duration compared to the duration of the cognitive tasks to which subjects are commonly exposed. Therefore, mental effort was not to be considered extreme in present study, since higher psychobiological stress levels (i.e. mood disorders, higher than normal perception of effort during training and reduced performance) can induce symptoms of mental fatigue even in elite athletes [45]. On the other hand, longitudinal studies on associations between mental fatigue and physical performance are scarce. It is unknown how longitudinal changes in mental fatigue are associated with changes in performance. Although we can assume that increasing the fatigue times could induce the improvement in reaction time (for explample during the Stroop task), due to the habituation effect.”
Round 2
Reviewer 1 Report (New Reviewer)
I would like to congratulate the authors for the effort made in the development of this article.
Reviewer 2 Report (New Reviewer)
I would like to congratulate with the authors. The changes provided are helping the reader to have a better understanding of the experiment and findings.
This manuscript is a resubmission of an earlier submission. The following is a list of the peer review reports and author responses from that submission.
Round 1
Reviewer 1 Report
Dear Authors,
The article and research are interesting areas of science. However, I have some comments and questions regarding the research methodology and, above all, the conclusions and their applications. My comments are below.
In my opinion, lactatemia is incorrectly indicated here, as is metabolic acidosis. These are pathologies that last over time. In the event of exercise, there is said to be a high concentration of lactate, which is removed, for example, by lactate resynthesis. For example, blood buffers and the availability of oxygen after exercise effectively protect against the high concentration of hydrogen ions. The issues indicated in the text concern diseases, not physical exercise athletes (line 48).
Were the subjects' diets monitored? If not-why?
Were there any exclusion criteria?
What physical activity did the respondents perform during the week? What were her character, type, and intensity? How was it monitored? How was it rated? Were they highly trained or not? This affects their body's assessment of this effort.
Why was it not assessed in which exercise zones the subjects pedaled? Was it moderate, low, or high intensity? This significantly affects the energy of effort and evaluation. , pedaling with an effort perception produced and imposed by the subjects themselves with reference’’. If the measured gas exchange can be assessed in some energy zones the effort was made.
Did the respondents consume fluids, vitamins, minerals, and carbohydrates during exercise?
Was the study conducted on an empty stomach?
In my opinion, the discussion should be deepened. A reference to other authors in many places would increase the credibility of the conclusions.
It is worth adding the limitations of the study (e.g. a small number of respondents, which is immediately noticed).
20 minutes of cycling, of unknown intensity, is unfortunately not a great challenge even for amateurs. The assessment of the sports level of people and the whole group (differentiation) is crucial here.
It should be indicated in the text how much glucose the brain uses during cognitive work. It is also important to indicate the reserves of glucose, ATP, phosphocreatine, and glycogen in the liver and muscles.
This is the basis of this article. It is important to show how much the brain can use glucose for different activities, according to the literature.
The first conclusion is indicated in the conclusions. So where are the next papers in the papers?
This conclusion is somewhat contradictory and confusing. Mental attitude and mental fatigue can influence, and at the same time does not?
This knowledge, especially of the nature of the research effort, can be proposed to amateurs. Certainly not to professional athletes. In the latter group, measurable measurements are used to assess the severity of the effort. Brain disorders may affect, for example, the motivation to exercise. However, to achieve the best results in sport, it is necessary to make the effort anyway. Sometimes
the perception of exercise is more severe in subjective assessment than objective (measuring lactate during exercise and assessing exercise zones). It is also influenced by the concentration of glutathione, glutamine, and the dependence of testosterone on cortisol.
These are indicators used in competitive sports to assess effort or potential overtraining. The RPE scale is not very precise and can only be potentially used by coaches. The more detailed ones are indicated above. However, such activities can be used by amateurs or physically active people. And it offers potential opportunities to use this knowledge. However, the study group should be precisely defined, its training experience, the level of efficiency, the training performed, and the assessment of the severity of the examination from the perspective of physiology.
Kind regards,
Reviewer